# Renewable Energy Communities as Modes of Collective Prosumership: A Multi-Disciplinary Assessment Part II—Case Study

Shubhra Chaudhry [1,†], Arne Surmann [2,*,†], Matthias Kühnbach [2] and Frank Pierie [1]

1  Institute of Engineering, Hanze University of Applied Sciences, 9747 AS Groningen, The Netherlands
2  Department of Smart Grids, Fraunhofer Institute for Solar Energy Systems ISE, 79110 Freiburg, Germany
*  Correspondence: arne.surmann@ise.fraunhofer.de; Tel.: +49-761-4588-2225
†  These authors contributed equally to this work.

**Abstract:** Renewable Energy Communities (RECs) have been defined as modes of collective prosumership under the Renewable Energy Directive (RED II). We evaluate the benefits offered by RECs and the barriers and enablers impacting their uptake. Germany is taken as a case study for a novel multi-disciplinary assessment of a potential REC intended as a climate-neutral, mixed-use district. We found that energy cooperatives may not be suited to form RECs, but the future may see an uptake of innovative organizational structures such as the Consumer Stock Ownership Plan. It has been shown that a high degree of prosumership can provide technical and economic benefits with maximum greenhouse gas savings of 35% and a maximum self-consumption share of 61% compared to no prosumership. The REC has a negative Net Present Value (NPV) after 25 years of operation and lacks financial attractiveness. A positive NPV is only possible by using the cost savings from prosumership to recoup the investments faster. RECs are a promising mode of citizen participation in the energy transition; however, for their application in Germany, together with the currently missing regulatory allowance of sharing energy between small-scale parties over a public grid, dedicated subsidies, one-time grants or price support for operators are needed.

**Keywords:** Renewable Energy Communities; prosumer; Renewable Energy Directive; Consumer Stock Ownership Plan (CSOP); Tenant Electricity Model

## 1. Introduction

The emergence of local energy systems in Europe is changing who owns, generates and distributes energy. Citizens are transitioning away from being passive consumers and taking an active role as prosumers and co-owners of distributed energy systems [1]. Active participation in renewable energy and prosumership are cornerstones of a successful energy transition. Community Renewable Energy (RE) initiatives offer multitudes of benefits—they democratize the energy transition [2,3], maximize the consumption of locally produced clean energy at lower prices, and can reduce grid stress by aggregating the potential of individuals to offer demand side management.

Historically speaking, community energy initiatives have displayed immense diversity—in terms of the organizational structures used, technologies deployed and areas of activities [2]. The European Union (EU) attempted to homogenize this diversity and offer a regulatory framework for such initiatives, leading to the two definitions of Energy Communities—Renewable Energy Communities (RECs) under the Renewable Energy Directive (RED II) [4] and Citizen Energy Communities (CECs) under the Internal Electricity Market Directive [5]. This article focuses on RECs which are entities entitled to produce, consume, store, sell and share renewable energy. RECs can participate in energy markets directly or through aggregators and can enjoy favorable conditions under enabling frameworks to be developed by all Member States [4].

Germany offers ideal conditions as a 'regulatory sandbox' to test the next stage of growth of community energy projects under the new regulatory frameworks. It has been a pioneer of the European energy transition since the 1990s and has a strong movement of community energy projects due to a favorable regulatory landscape and intrinsically motivated citizens [6–8]. In 2019, Germany had over 1700 community energy projects,—the highest in the EU [2,9], and these initiatives collectively formed the 14th largest energy retailer in Europe in 2016 [10]. The country also offers untapped technical potential for prosumership: 3.8 million apartments are found to be suitable for PV installations [11] and the unused rooftop area suitable for PV installations is estimated to be 89%; the potential for facade installations is even higher [12].

In terms of regulatory provisions, Germany has not fully transposed the latest directive on RECs in the national laws. Currently, the possibility for RECs in Germany is under the Tenant Electricity Model (Mieterstrommodel) within the Renewable Energy Sources Act 2021 (EEG 2021) [13]. Under this model, landlords or third parties can install PV panels in the multi-apartment buildings (with greater than 40% area dedicated to residential uses) and supply the tenants with the electricity generated [11]. The landlord is free to supply the excess electricity to residential or ancillary buildings in the neighborhood, provided the public grid is not used or can feed it to the grid [13,14]. To incentivize participation in the model, the self-consumed electricity is exempt from all surcharges and the landlords receive a bonus (called Landlord Bonus or Tenant Electricity Surcharge) to compensate for the additional administrative costs. The Tenant Electricity Model can technically produce 14 TWh of additional renewable electricity across 3.8 million apartments [11]. Despite the high technical potential and availability of incentives, only 1% of the budget for the Tenant Electricity Model had been used up by 2019 and 30 MW of systems installed, remaining significantly below targets [11,15]. Since tenants do not own any RES, do not interact with each other and remain passive consumers within this model, it does not comply with the RED II. However, it is worth testing the provisions on a potential REC to explore the reasons for the slow uptake of Tenant Electricity.

Community energy initiatives and collective prosumership have been well-researched in the scientific literature; however, these studies tend to focus on specific disciplines, e.g., energy, economics, regulatory, or social. Caramizaru and Uihlien [2] studied the regulatory provisions for RECs and identified the benefits of RECs for stakeholders and the energy system. Lowitzsch et al. [16] studied the regulatory and governance criteria for RECs; and found that only 9 of the 67 met all the governance criteria of RED II and recommended that 'enabling frameworks' be designed to encourage the inclusion of a variety of actors and RE technologies in RECs. Guidelines to develop such enabling frameworks were offered in [17] with a special focus on the inclusion of low-income consumers in the collective prosumership project. Fleischacker et al. [18] researched the economic and ecological impacts of forming an energy community in Austria and concluded that forming energy communities could lead to a reduction in system costs and Greenhouse Gas (GHG) emissions but these were inversely related, that is, minimum systems costs were accompanied by maximum GHG emissions and vice versa. Moncecchi et al. [19] developed a techno-economic model for energy sharing within an energy community in Italy and assessed the overall feasibility of the project and the impact of financial incentives on the Net Present Value (NPV) of the project. They concluded that systems based on only energy production had higher NPV than those based on self-consumption and energy sharing. Azarova et al. [20] looked at energy communities in Germany, Austria, Italy and Switzerland from a social perspective to conclude that choice of RE influenced the social acceptance of initiatives, with energy communities based on PV and power-to-gas technologies enjoying higher acceptance across the nations compared to other technologies. Horstink et al. [21] studied organizational structures of community energy projects across nine European nations to find that the selection of the organizational structure was based on the available legal forms and access to support and subsidy schemes in the country. Lowitzsch [16,22,23] looked at the organizational and financial aspects of RECs in the

Czech Republic, Germany and Poland to demonstrate the Consumer Stock Ownership Plan (CSOP) as an emerging but viable model for tenants to collectively invest in and co-own renewable energy and energy efficiency projects.

While energy communities and adjacent topics have been extensively researched across one or two disciplines as discussed previously, integrated and multi-disciplinary studies on RECs are lacking. We address this gap by assessing a potential REC in Germany across multiple disciplines including regulatory, organizational, technical, economic, and ecological aspects. A mixed-use neighborhood which is currently under construction within the German city of Kaiserslautern is used as potential REC to answer the following research questions (RQs):

**RQ1** How could a Renewable Energy Community (REC) be organized and financed to be compliant with the governance criteria set out in RED II?

**RQ2** What are the benefits of forming a REC and is such a project economically feasible?

**RQ3** Which regulatory provisions act as enablers for the development of RECs and which act as barriers?

*Paper Structure*

This article is the second in a two-part paper on RECs as modes of collective prosumership. In the first part [24], a replicable model for assessing RECs was introduced. The model offers an evaluation suite from a technical, economic and ecological perspective. In this article, the assessment framework, including the modeling method, is applied to a real-world German demonstration district. The study is organized as follows: Section 2 describes the methods used for the multi-disciplinary assessment of RECs and the framework and modeling method applied. Section 3 gives details of the potential REC used as a case study while Section 4 describes the different use cases and scenarios of prosumership analyzed. Section 5 presents the results of the assessment of the potential REC. Section 6 discusses the regulatory barriers and enablers for RECs and reflects on the limitations and areas of future research. Section 7 concludes the study.

## 2. Methodology

The framework to undertake the multi-disciplinary assessment of RECs (depicted in Figure 1) aims to understand the impact of regulatory provisions (such as regulations, taxes and incentives) on the organizational, technical, economic, and ecological aspects of RECs [25]. The organizational aspects for RECs may include organizational structures and their impact on members; the technical aspects include the energy flows in decentralized energy systems; the ecological aspects are represented by emissions of greenhouse gases (GHG); and the economic aspects include the energy costs and financial profitability of the REC [25]. Social aspects, such as stakeholder engagement and social acceptance of the project, are beyond the scope of this article.

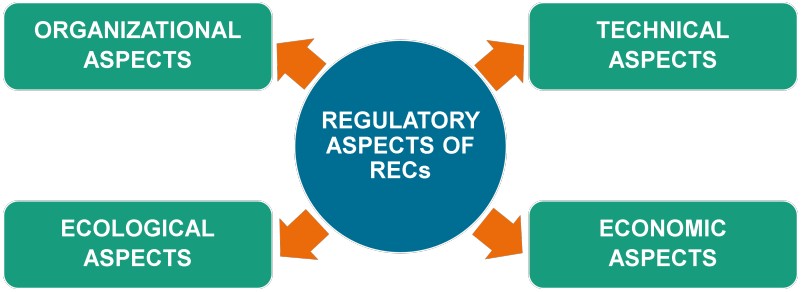

**Figure 1.** Aspects analyzed within this study following the PESTEL framework.

### 2.1. Organizational Aspects of RECs

Community energy projects have been organized in a variety of legal/organizational structures, but it is unclear whether these forms are compliant with RED II to form RECs.

Therefore, to answer RQ 1, traditionally prevalent and emerging organizational structures were evaluated against RED II. For a community energy project to be recognized as a REC and to benefit from the enabling framework under RED II, the project must have an organizational structure that meets the criteria set out in RED II, that is, it must be autonomous; effectively controlled by members who are in the proximity of the project; should involve heterogeneous actors such as citizens, municipalities or Small and Medium Enterprises (SMEs) as its members and co-investors; and have no individual shareholder owning more than 33% shares [4].

In Germany, energy cooperatives have historically been an immensely popular form of organizing community energy projects. Over half of all German community energy projects in 2019 were energy cooperatives [2,9,26]. They have enabled motivated citizens to pool their savings or disposable incomes to collectively invest in RE projects. They enable democratic decision making through the 'one-member, one-vote' principle, irrespective of the shares held by the member. However, cooperatives suffer inherent disadvantages that make them unsuitable to form a RED II-compliant REC. Firstly, the 'one-member, one-vote' principle discourages membership in the REC by professional actors who prefer voting rights in proportion to their shareholding [22]. Secondly, citizens must contribute financially to become members of cooperatives which leads to the exclusion of poor citizens and contradicts the very essence of RED II. Thirdly, the scale of investments possible is limited by the disposable incomes of the members, as a result, cooperatives are not suited to larger investments. It can be concluded that going forward, cooperatives may lose popularity when forming RECs.

Innovation in organizational structures is needed to form RED II-compliant RECs. One such innovation is the Consumer Stock Ownership Plan (CSOP) applied to renewable energy (RE-CSOP).

A CSOP is defined as an organizational structure that "enables consumers—especially those without savings or access to capital credit—to acquire an ownership stake in a utility that supplies them, and thus, to become 'prosumers'" [22]. It is a low-threshold financing model that does not rely on the monetary contributions from members by raising capital from external sources against future earnings from the RE project, thus opening the membership to all citizens and enabling a larger scale of investments. CSOPs facilitate co-ownership by diverse actors such as municipalities, SMEs, local partners or commercial investors. It pools the voting rights of citizen co-owners in proportion to their shareholding, thus removing the disadvantage of cooperatives to other co-investors while still ensuring a democratic decision-making process [16,22,23] . RE-CSOPs comply with RED II, address the disadvantages of cooperatives and offer additional benefits to members and can therefore be used to form the REC under study.

### 2.2. Technical, Ecological and Economic Aspects of REC

The integrated methodology to assess the technical, ecological and economic aspects of a REC follows the REC Assessment Model introduced in part one of this paper series [24]. The following three sub-chapters, therefore, briefly summarize the key expressions needed to understand the case study results while an extensive description is given in [24].

### 2.3. Technical Assessment

The Technical Assessment module (derived from [24], Section 2.5) undertakes an energy flow analysis for the decentralized energy system of the REC. The energy demand is calculated based on the annual load profiles of buildings in the REC and the energy generated is calculated from the annual PV generation profiles. If self-consumption is pursued, then the PV energy generated in a building is used to meet the instantaneous demand of the building. If energy sharing within the REC is pursued, then the surplus energy in a building is supplied to other buildings in the REC to meet their energy demand. Any remaining energy is supplied to the grid. Any demand deficit that cannot be met by self-consumption or from energy shared in the REC is fulfilled through electricity from the grid.

Considering a REC of $N$ buildings, with each building $n$ having an energy load $E_{load}$ and generating solar Photovoltaic (PV) energy $E_{pv}$ at time $t$. If the building self-consumes $E_{self-cons}$ and consumes $E_{shared}^{rec \rightarrow b}$ from the energy shared in the REC, then,

- The self-sufficiency share ($SSS$) of the REC is a metric to measure the proportion of energy demand of the entire REC that can be met through energy generated in the REC, expressed as:

$$SSS = \frac{\sum_{n=1}^{N} \sum_{t=1}^{T} E_{self-cons}(t,n) + E_{shared}^{rec \rightarrow b}(t,n)}{\sum_{n=1}^{N} \sum_{t=1}^{T} E_{load}(t,n)} \tag{1}$$

- The PV self-consumption share ($SCS$) can be used to measure the amount of total PV generated in the REC which is consumed within the REC itself, calculated as:

$$SCS = \frac{\sum_{n=1}^{N} \sum_{t=1}^{T} E_{self-cons}(t,n) + E_{shared}^{rec \rightarrow b}(t,n)}{\sum_{n=1}^{N} \sum_{t=1}^{T} E_{pv}(t,n)} \tag{2}$$

*2.4. Ecological Assessment*

The Ecological Assessment module (derived from [24], Section 2.6) calculates the Greenhouse Gas (GHG) emissions of energy consumption. The module includes the direct operational emissions resulting from energy generation and omits the indirect emissions resulting from construction, fuel provision or decommissioning of energy generators. The GHG emissions of the REC, expressed in carbon dioxide equivalents ($CO_2eq$), can be calculated as:

$$GHG = E_{load}^{grid} * F_{GHG}^{grid} + (E_{self-cons} + E_{shared}^{rec \rightarrow b}) * F_{GHG}^{pv} \tag{3}$$

where $E_{load}^{grid}$ is the energy consumed by the REC from the grid with a grid GHG emission factor of $F_{GHG}^{grid}$ while $F_{GHG}^{pv}$ is the GHG emission factor of PV energy.

*2.5. Economic Assessment*

The Economic Assessment module (derived from [24], Section 2.7) looks at two perspectives. The annual costs of energy consumption in the REC are calculated from the consumers' perspective while the Net Present Value (NPV) is calculated to evaluate the profitability of the project from the investors' perspective. The cost of energy consumption is calculated based on the applicable electricity tariff and taxes. The NPV is calculated by discounting the project cash flows of the REC over its lifetime. A project with a positive NPV is accepted for investment while one with a negative NPV is rejected.

- The total annual cost of energy consumption $C$ is calculated as a sum of its components, namely, grid usage costs ($C_g$), fixed supply costs ($C_{sf}$), variable supply costs ($C_{sv}$), energy taxes ($C_{tax}$) and retailer margin ($C_{rm}$):

$$C = C_g + C_{sf} + C_{sv} + C_{tax} + C_{rm} \tag{4}$$

- The NPV of the project is calculated by discounting the cash flows $CF$ of the project for year $y$ over the project lifetime $T$ using a discount rate $d$ that has been adjusted for inflation:

$$NPV = \sum_{y=1}^{T} \frac{CF(y)}{(1+d)^y} \tag{5}$$

## 3. Case Study

For this study, we selected a German neighborhood within a real-world laboratory in the ongoing research project called 'EnStadt:Pfaff' [27]. The Pfaff neighborhood is characterized by the coupling of multiple energy sectors—electricity, heating and mobility—and a close proximity of all renewable energy sources (RES) and consumers, making

it suitable to be studied as a potential REC. The neighborhood will have 27 buildings consisting of 3492 units which will be rented out for residential, office or commercial activities. All buildings will be equipped with PV panels on their rooftops and facades with a total installed capacity of 5.3 MW. Some residents are assumed to own Electric Vehicles (EVs). Table 1 provides further details about the neighborhood.

**Table 1.** Details of the studied neighborhood.

| Building | | | | | | Electric Vehicle | |
|---|---|---|---|---|---|---|---|
| Number | Name | Type | Number of Rental Units | PV Installed Capacity | Annual Consumption | Number of Vehicles | Annual Consumption |
| - | - | - | - | kWp | MWh | - | MWh |
| 1 | MU 1.1 | Office | 376 | 416 | 563 | 57 | 27 |
| 2 | MU 1.2 | Office | 91 | 153 | 137 | 14 | 9 |
| 3 | MU 1.3 | Office | 107 | 129 | 160 | 16 | 9 |
| 4 | MU 1.4 | Residential | 483 | 290 | 477 | 53 | 36 |
| 5 | MU 2.1 | Residential | 180 | 152 | 165 | 20 | 13 |
| 6 | MU 2.2 | Residential | 178 | 218 | 167 | 20 | 14 |
| 7 | MU 2.3 | Office | 90 | 119 | 135 | 14 | 11 |
| 8 | MU 3.1 | Commercial | 12 | 118 | 68 | 0 | 0 |
| 9 | MU 3.2 | Office | 100 | 124 | 150 | 15 | 10 |
| 10 | MU 3.3 | Commercial | 10 | 107 | 30 | 0 | 0 |
| 11 | SO 1.1 East | Office | 106 | 99 | 158 | 16 | 10 |
| 12 | SO 1.1 West | Office | 151 | 181 | 226 | 23 | 14 |
| 13 | SO 1.2 | Office | 33 | 116 | 49 | 5 | 2 |
| 14 | SO 1.3a | Office | 65 | 98 | 98 | 10 | 6 |
| 15 | SO 1.3b | Office | 9 | 33 | 13 | 1 | 1 |
| 16 | SO 1.4 | Office | 10 | 28 | 15 | 1 | 0 |
| 17 | SO 2a | Office | 159 | 271 | 942 | 24 | 15 |
| 18 | SO 2b * | Office | 0 | 65 | 53 | 412 | 33 |
| 19 | SO 2c | Office | 22 | 255 | 64 | 3 | 1 |
| 20 | SO 3a | Commercial | 130 | 273 | 404 | 0 | 0 |
| 21 | SO 3b * | Office | 0 | 197 | 36 | 412 | 34 |
| 22 | SO 4.1 | Office | 194 | 244 | 807 | 29 | 13 |
| 23 | SO 4.2 | Office | 246 | 198 | 368 | 37 | 19 |
| 24 | SO 5.1 | Commercial | 98 | 256 | 200 | 0 | 0 |
| 25 | SO 5.2 | Office | 233 | 339 | 350 | 35 | 20 |
| 26 | SO 5.3 | Office | 318 | 418 | 476 | 48 | 27 |
| 27 | SO 6.1 | Office | 92 | 399 | 190 | 14 | 10 |
| | | Office | 2402 | 3882 | 4990 | 1186 | 271 |
| | Total | Commercial | 250 | 754 | 702 | 0 | 0 |
| | | Residential | 841 | 660 | 809 | 93 | 63 |
| | | All | 3493 | 5296 | 6501 | 1279 | 334 |

* These buildings offer parking facilities to offices and are classified as Office buildings. These two parking facilities are proposed to be the sites of the public charging infrastructure for electric vehicles.

### 3.1. Actors within REC Pfaff

Four actors have been identified for REC Pfaff: the tenants who consume electricity; the landlords of the buildings; the Municipal Electricity Utility which retails electricity to the area; and finally, a (development) bank that offers an investment loan. Since each actor has different objectives, a set of Key Performance Indicators (KPIs) have been mapped to them based on expressions derived from the REC Assessment Model [24]. Table 2 lists the roles played by the various actors, their objectives for joining the REC and the corresponding KPI.

**Table 2.** Roles, Objectives and KPIs for Actors of REC Pfaff.

| | Tenants | Landlords | (Development) Bank | Municipal Utility |
|---|---|---|---|---|
| **Role** | • Prosumer of el. | • Allows installation of PV systems on buildings | • Provider of loan to set up REC | • Installation, O&M of PV<br>• Energy retailer to consumers<br>• Operator of EMS platform<br>• Balance Responsible Party |
| | | • For all: co-ownership of REC | | |
| **Objective** | • Lower prices<br>• Consumption of local green energy<br>• Reduction in GHG footprint | • Fulfilling building standards to dedicate rooftop area to PV systems<br>• Increasing the real estate value | • Funding the federal climate and energy goals by offering credit to projects that benefit the environment and society<br>• Improved branding and reduced Environment, Social, Governance risks | • Opening a new revenue stream as REC operator and service provider<br>• Scaling up of RE production<br>• Improved branding as a sustainable business |
| | | • For all: active participation in the energy transition | | |
| **KPI** | • Lowest cost of el. (Equation (4))<br>• Highest SSS (Equation (1))<br>• Lowest GHG (Equation (3)) | • Highest NPV (Equation (5)) | • Highest NPV (Equation (5)) | • Highest SCS (Equation (2)) |

### 3.2. Simulation Setup for the Application of REC Assessment Model

The load and production profiles representing the demonstration neighborhood were generated using the synPRO stochastic model [28] based on factors such as the number of consumers in a household/unit, their socioeconomic status, the electrical appliances used by them and the efficiency of each appliance [29]. In addition, two sets of load profiles for the electric vehicles were generated using the synPRO-EV sub-model [30]. The first set follows a "charge full upon arrival" logic while the second applies controlled charging to maximize PV self-consumption to offer demand response through implicit flexibility [31]. The models were used to generate load profiles for buildings and EVs for one year with a time interval of 15 min. The time series generated from the models and the inputs presented in Table 3 were used to undertake simulations with the REC Assessment Model. Information on electricity price assumptions is listed in Appendix Table A1 together with the feed-in and tenant electricity remuneration Table A2.

**Table 3.** Inputs to the REC Assessment Model.

| Parameter | Value | Unit | Source |
|---|---|---|---|
| **Technical Analysis** | | | |
| PV installed capacity | 5296 | kWp | Project Data |
| No. of buildings (N) | 27 | | Project Data |
| Time interval ($\Delta t$) | 15 | minutes | Input from synPRO [29,30] |
| Share of el. vehicles | 30% | | A realistic scenario for vehicle electrification in Kaiserslautern by 2030 [32] |
| **Ecological Analysis** | | | |
| GHG emission factor for grid el. in Germany | 0.33866 | $\frac{kgCO_2eq}{kWh}$ | Emission factor based on the mix of fuels used by power stations to generate electricity in Germany in 2021, from EcoInvent 2.2 based on GWP100a [33] |
| GHG emission factor for self-generated PV | 0 | $\frac{kgCO_2eq}{kWh}$ | Assuming negligible operational GHG emissions from grid-connected PV systems [34] |
| **NPV Analysis** | | | |
| Discount rate (d) | 5% | | [35] |
| Expected lifetime PV system | 25 | years | [35] |
| CAPEX (PV system) | 800 | $\frac{€}{kWp}$ | High-cost estimate for utility scale PV systems [36] |
| OPEX (PV system) | 800 | $\frac{€}{a*kW}$ | [35] |
| OPEX (REC) | 22 | $\frac{€}{a*kW}$ | Assuming that OPEX of REC is 1.1× the OPEX of PV system |
| Equity | 10% | | Assuming each rental unit contributes 120€ as one-time contribution to equity |
| Loan repayment period | 10 | years | Terms of loan by KFW Bank [37] |
| Interest rate | 2.46% | | Terms of loan by KFW Bank [37]] |
| Effective corporate tax rate | 30.2% | | Tax rate for Kaiserslautern [38,39] |
| Inflation rate | 1.27% | | Average from 2011–2020 [40] |
| Incentive duration | 20 | years | EEG 2021 [13] |
| Electricity price at the end of incentive period | 0.083 | $\frac{€_{2015}}{kWh}$ | Assuming high wholesale electricity price development [41] |

## 4. Use Cases and Scenarios

Different combinations of activities give rise to different degrees of prosumership that can be pursued in a REC. Figure 2 pictorially represents the use cases and scenarios that were modeled to evaluate the impact of different levels of prosumership on the outcomes of the REC Assessment Model followed by a description of the use cases:

| Activity Pursued | Use Case → | Base Case | No Prosumership | | Partial Prosumership | | Tenant Electricity Model | | Full Prosumership | |
|---|---|---|---|---|---|---|---|---|---|---|
| | Scenario → | Base | NP | NP_flex | PP | PP_flex | TE | TE_flex | FP | FP_flex |
| Creation of REC | | | ■ | ■ | ■ | ■ | ■ | ■ | ■ | ■ |
| PV Generation | | | ■ | ■ | ■ | ■ | ■ | ■ | ■ | ■ |
| PV Feed-in | | | ■ | ■ | ■ | ■ | ■ | ■ | ■ | ■ |
| Self-consumption | | | | | ■ | ■ | ■ | ■ | ■ | ■ |
| Energy Sharing | | | | | | | | | ■ | ■ |
| EV Flexibility | | | | ■ | | ■ | | ■ | | ■ |

**Figure 2.** Use Cases and scenarios based on different degrees of prosumership pursued in a REC. The cells shaded in green imply that the activity has been pursued while empty cells imply that the activity has not been pursued.

- Base: This use case forms the baseline of the analysis and represents the scenario in which the neighborhood does not collectively organize into a REC and does not generate any renewable energy.
- No Prosumership (NP): In this use case, the neighborhood forms a REC and generates PV energy from its owned assets. However, it does not pursue prosumership by consuming any of its self-generated energy. Instead, all the energy produced is fed into the grid and compensated by the full feed-in tariff [42]. The energy needs of the community are met through grid-supplied electricity.
- Partial Prosumership (PP): In this use case, the REC is formed, and the PV generated is used for self-consumption. Any surplus after self-consumption within the building is fed into the grid and financially compensated with the surplus feed-in tariff [42]. Periods of demand deficit are met with grid electricity.
- Tenant Electricity Model (TE): This use case represents the possibilities of prosumership under the Tenant Electricity Model under the EEG 2021. Here, the REC is formed, and PV is generated. Self-consumption occurs in all buildings. The law dictates that energy can be shared from a residential building to another building in the same neighborhood if it is a residential or ancillary building [13]. Apart from three residential buildings, all other buildings are non-residential, so energy sharing within this REC cannot be pursued under the law. This use case can be seen as a modified Partial Prosumership case with an additional incentive (Tenant Electricity Surcharge).
- Full Prosumership (FP): This use case involves the full range of prosumership that can be undertaken at the REC. Here, the PV generated by the REC is first used for self-consumption within the building, then any surplus in a building is shared with the community to meet, fully or partially, the deficits occurring in other buildings of the REC. The remaining surplus energy is fed to the grid using the surplus feed-in tariff. Any remaining deficits are fulfilled through grid-supplied electricity.

The use cases in which a REC is formed are further divided into two scenarios: one without the application of flexibility and one, termed "flex", with demand side management where EV charging is shifted to periods of surplus PV generation.

## 5. Results

The results section presents the different dimensions of the REC Assessment model, starting with a general part on the organizational perspective (Section 5.1) and the overall energy demand (Section 5.2) of the demonstrated REC. It is followed by a scenario-specific view on the outcome of the simulated scenarios as described in (Section 3.2) with regard to energy self-consumption (Section 5.3), greenhouse gas reduction (Section 5.4), cost reductions from the consumer perspective (Section 5.5) and NPV from the REC-OC perspective (Section 5.7).

### 5.1. Organizational Structure

The organizational structure of the REC imagined within the EnStadt:Pfaff project is designed using the RE-CSOP due to its comparative advantages over traditional German energy cooperatives. The step-by-step formation of REC Pfaff is depicted in Figure 3. In the first step, all the consumers of the Pfaff neighborhood form a trusteeship and elect a trustee to be a shareholder on behalf of the consumers and to represent their interests in the REC. Each tenant of a rental unit contributes a small monetary amount (10% of the initial investment which equals 120€ in the specific case) as a one-time investment into the REC. The trustee, together with other co-investors (landlords, municipal utility, other SMEs), forms the REC Pfaff (Box in Step 2). The trustee and co-investors together set up an intermediate entity to operate the REC (Step 3). This "REC Pfaff Operating Company" (REC-OC) is owned by the investors in proportion to their shareholding. For illustration, it has been considered that the trustee uses the contributions from tenants to contribute 10% of the investment needed as equity while 90% of the investment is funded by a loan. The loan is attributed to the trustee and co-investors in a 60:40 ratio. As a result of this

setup, the tenants collectively own 64% of the REC and co-investors collectively own the remaining 36%, thereby fulfilling the criteria for autonomy and effective control as defined by the RED II.

The REC-OC raises a loan from the bank (Step 4) and invests it in PV systems on the buildings (Step 5). The REC-OC acts as a retailer and signs supplier contracts with the consumers in the buildings. It operates the PV assets and supplies PV-generated energy to the consumers (Step 6), who have now become prosumers, in return for a monthly tariff (Step 7). It sells any excess generated energy to the grid in return for a feed-in tariff or directly to the wholesale market once the period to avail of the feed-in incentive is over. Lastly, it purchases any energy needed to meet the demand that is not met through the PV.

The REC-OC uses the payment from selling the electricity to pay back the loan to the bank (Step 8). Once the loan has been repaid and the investment has recouped itself, the REC-OC can share any profits generated with the members of the REC, either as dividends (Step 9) or by further reducing the community electricity tariff. Other forms of returning the profits to the community can also be considered, for instance, using the profits to reinvest in solar panels reaching the end of life, to set up additional PV systems or even invest in energy efficiency to make the neighborhood even more sustainable and energy independent over time.

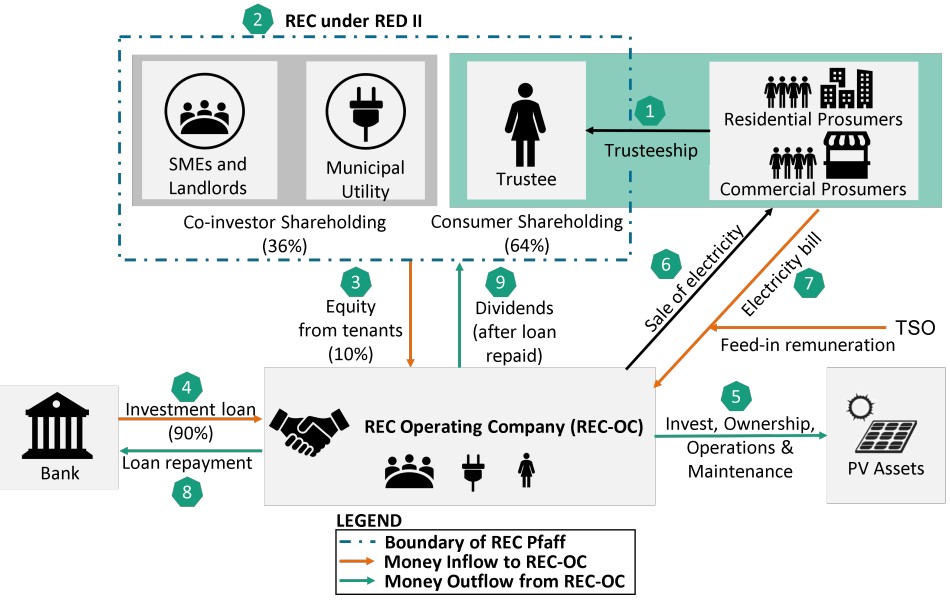

**Figure 3.** Organizational structure of REC Pfaff. The numbers represent the steps in the formation and operation of REC Pfaff.

This setup leads to a few other advantages beyond the ones mentioned in Section 2.1. Tenants may not want to invest in RE ventures because they may not accrue the benefits if they leave the neighborhood before the investment pays off. However, under this organizational structure, tenants can be members of the REC with minimal upfront investment and can easily transfer their shares within the trusteeship to the next tenant. A two-tiered structure means that the decision making is simple for non-prosumer co-owners (i.e., municipal utility and landlords) who only interact with the trustee (and not multiple prosumer co-owners) for day-to-day decisions. The REC enjoys collective leverage to raise one loan instead of multiple micro loans. Lastly, having the Municipal Utility as part of the REC-OC brings operational expertise to the REC [16]. For the (development) bank, the two-tier system is advantageous as it cuts transaction costs of offering loans. Instead of providing small loans to individuals, the banks are able to offer a single loan that is secured against the value of the PV assets and the future earnings of the project. The organizational design

of REC Pfaff using the RE-CSOP offers an innovative alternative for consumer co-owners to organize community energy projects in the future in full compliance with the RED II.

### 5.2. Energy Demand and Generation

The energy demand of REC Pfaff amounts to 6.8 GWh/a (see Table 1 total annual consumption including EVs). A total of 95% of this demand is attributed to the buildings (of which 77% is from offices, 12% from residential and 11% from commercial buildings) while 5% of the demand is for EV charging. REC Pfaff generates 3.9 GWh/a of PV energy. The community produces PV for 4300 h in a year with the PV generation exceeding the load (in kW terms) for 1700 h. Both the energy demand and the PV generation in the REC vary with the time of day and the season of the year. The daily and seasonal variability of energy demand and PV generation is shown in Figure 4.

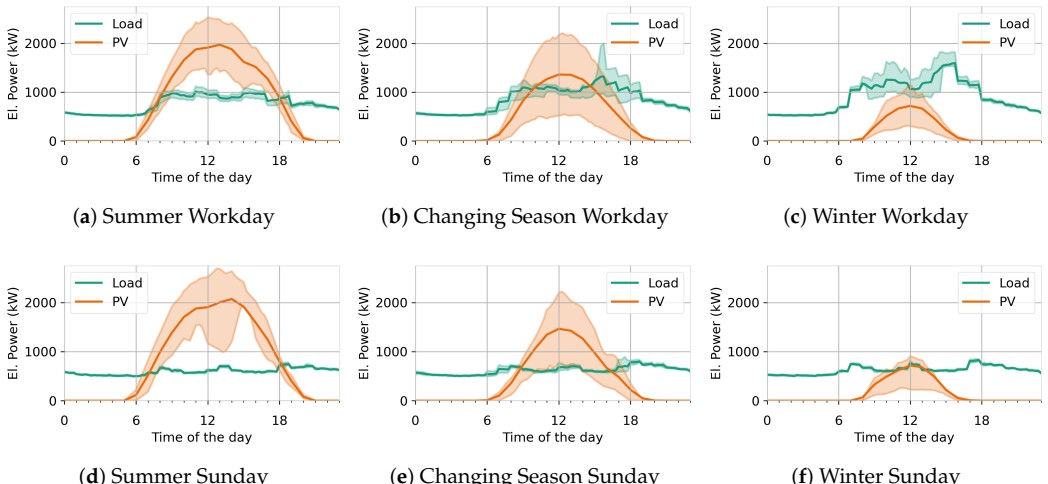

**Figure 4.** Temporal and seasonal variation in the electrical load (green) and PV generation (orange) in REC Pfaff with controlled charging applied during a mean workday and Sunday. The 25% and 75% percentiles are shaded. Seasons include: Summer (June–August), Changing Season (March–May, September–November), Winter (December–February).

One can see that the load matches consumption well during the workdays resulting in a self-consumption share of the REC (SSS) of 66% compared to 49% during Sundays. This is mainly due to the high number of offices and commercial units within the community. Working hours go well in line with daylight hours, which leads to a natural match of production and consumption without the need for energy management. During Sundays and public holidays, the energy consumption share by offices is reduced and the match is less pronounced. Naturally, during winter time, less PV production occurs, leading to a higher PV self-consumption share (+34% increase), while the self-sufficiency share is lower (−40% decrease) compared to the summer period. The seasonality of the load is less pronounced due to the installation of a high-temperature district heating system in the neighborhood which excludes the possibility of power-to-heat sector coupling for this particular community.

### 5.3. Increase in Self-Sufficiency Share (SSS) and Self-Consumption Share (SCS)

A major benefit of prosumership is that the REC can reduce its dependence on the grid. As Figure 5a shows, the Base and No Prosumership use cases have the lowest SSS (0%) and are fully reliant on the grid. The Full Prosumership use case leads to the highest SSS of 35.1% with flexibility, followed by 34.4% without flexibility. The Partial Prosumership and Tenant Electricity use cases offer an SSS of 28.8% with flexibility and 28.4% without flexibility.

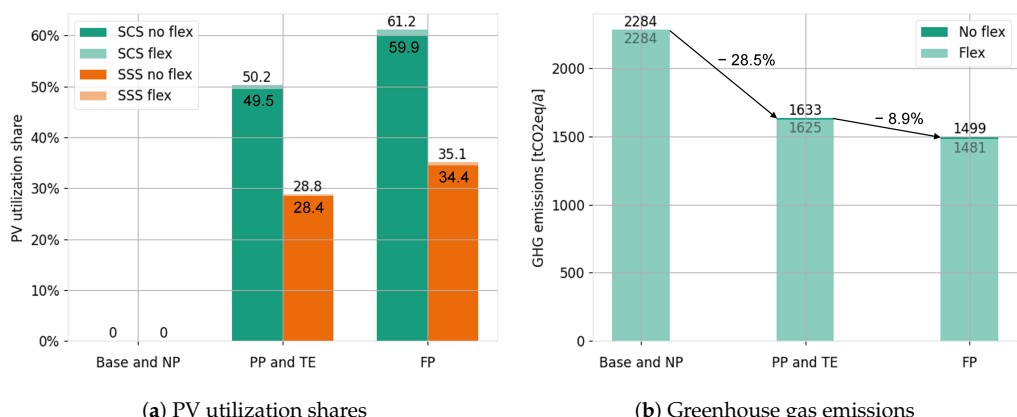

(**a**) PV utilization shares          (**b**) Greenhouse gas emissions

**Figure 5.** Impact of prosumership on PV Self-Consumption Share (SCS), Self-Sufficiency Share (SSS) (left) and greenhouse gas emissions (right). Some use cases have the same results and have been grouped together. The Base and No Prosumership (NP) are one group, and the Partial Prosumership (PP) and Tenant Electricity (TE) forms another group. Lighter hues represent the application of flexibility.

A similar trend is seen in SCS: the Full Prosumership use case makes the highest utilization of locally generated PV energy, having an SCS of 61.2% when flexibility is offered and 60% when no flexibility is offered. The Partial Prosumership use cases (including the Tenant Electricity Model) lead to an SCS of 49.5–50.2%, with the excess energy being supplied to the grid. The Base and No Prosumership use cases do not consume any of the PV energy generated and have an SCS of 0%. The flexibility offered through demand response has only a minor impact on the SSS and SCS of REC Pfaff since EV charging only accounts for 5% of the overall load.

With regard to the KPIs of the different actors: for the electricity utility, half the PV energy can be sold locally under Partial Prosumership. An additional 10% can be sold locally through energy sharing, making this the most attractive option. For the tenants', the Full Prosumership is the most attractive option, by supplying 35% of their demand themselves.

### 5.4. Reduction in GHG Emissions

By pursuing prosumership and consuming locally produced GHG-neutral energy, REC Pfaff can avoid GHG emissions that would have resulted from consuming GHG-intensive electricity from the grid. Figure 5b shows that the use cases without prosumership (Base and NP) have the highest GHG emissions of 2284 tons CO2eq per year. Pursuing Partial Prosumership under PP and TE use cases can reduce GHG emissions by 28.5%. Full Prosumership further cuts the GHG emissions down by 8.9%, offering an overall reduction of 35.2% reduction over the Base/NP case when offering flexibility and 34.3% without flexibility.

It is worth noting that REC Pfaff feeds excess PV energy to the grid, which can lead to the overall reduction in the GHG emissions attributed to the German electricity grid. This impact from a single REC is negligible and not quantified but as RECs and other decentralized energy systems based on RES proliferate in the coming decades, the GHG emission factor of the German electricity grid will also decrease, making grid electricity less GHG-intensive. Overall, prosumership facilitates actual reduction in GHG emissions from electricity consumption.

### 5.5. Economic Viability: Overview on Money Flows

Before investigating the economic impact of REC Pfaff from a consumer (Section 5.6) and an investors (REC-OC) view (Section 5.7), a general overview of money flows and prices are given. In Figure 6, the energy supply chain for 100 kWh of electricity is presented

in green and the money flow in exchange for those 100 kWh is visualized in blue based on the total annual costs. For 100 KWh demand in the REC, 28.9 Kwh are self-consumed and 6.31 kWh are consumed from shared energy. The remaining demand of 64.87 kWh is supplied by the grid. Excess energy generated in the REC, amounting to 22.26 kWh is fed-in into the grid. For this 100 kWh consumption, the tenants pay 27.56 € to the REC-OC which distributes the money further: 3.16 € go to the DSO for grid usage and 19.54 € to the municipal utility for energy supply. After 4.06 € as taxes, the REC-OC ends up with a retailer margin of 0.79 €. In addition, the 22 kWh fed into the grid are remunerated with 1.35 € as an income of REC-OC.

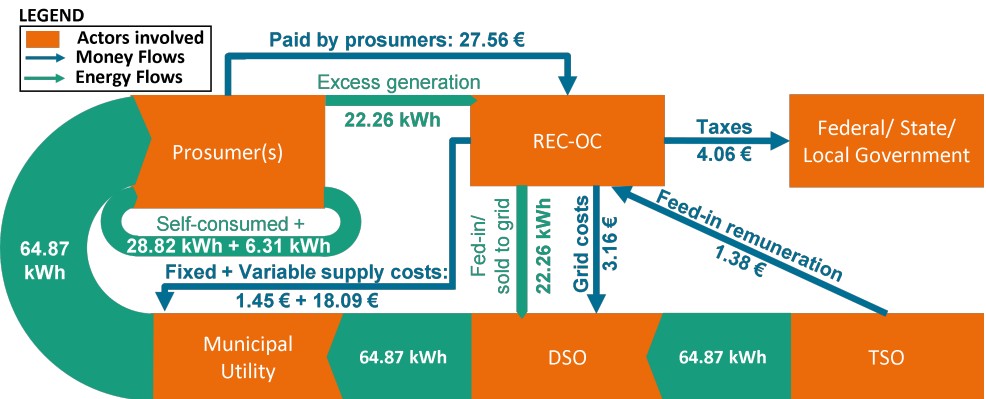

**Figure 6.** Energy and money flows between the various actors standardized for 100 kWh consumption. Exemplary for the full prosumership flexibility (FP_flex) scenario.

### 5.6. Economic Viability: Cost Reduction for Consumers

An economic perspective for the consumers within REC-Pfaff provides useful insights to answer the question of whether the founding of REC makes sense from a financial view. Prosumership brings down the overall cost of energy consumption for prosumers, partly due to the reduced demand for grid electricity and partly due to the reduction in taxes and levies which are provided as incentives to encourage self-consumption of energy (see Table A1). Figure 7 shows the breakdown of energy cost under various use cases in descending order. The data includes fixed-cost components (€/a) and consumption-dependent components (€/kWh). To make the data comparable, all costs are summed up over the period of one year and divided by the annual consumption.

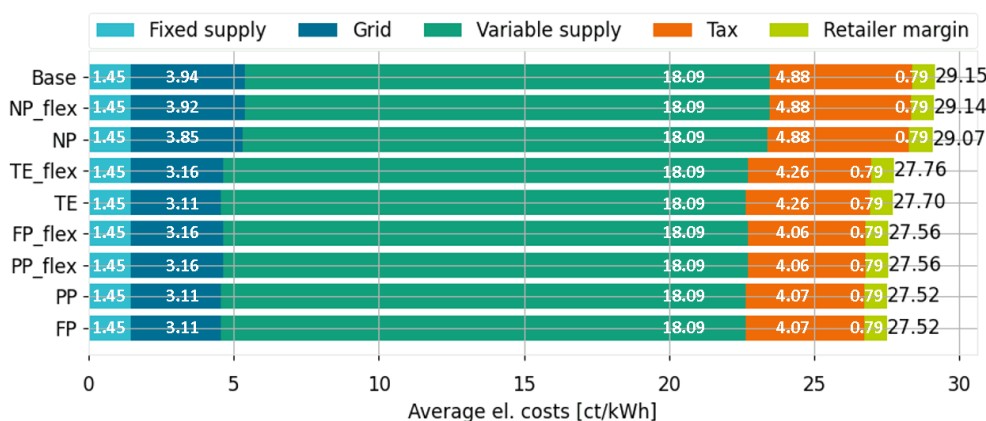

**Figure 7.** Cost of energy consumption for consumers under various use cases of prosumership. Values are calculated based on the annual costs for the different price components divided by the total annual energy consumption.

To showcase the cost savings, all the benefits of reduced costs are passed on to the prosumers. The higher the level of prosumership, the lower the cost of energy consumption. The average electricity price for the base case scenario is 29.15 ct/kWh. It decreases by 4% in the Tenant Electricity use case and 5% when Partial or Full Prosumership is followed. In all cases with prosumership, the root cause of the decline in energy costs is the tax component. This is because the self-consumption of energy is exempt from all taxes and surcharges. Further, energy consumed from the grid declines when going from No Prosumership to Full Prosumership. Due to the combined effect of reduced taxes and lower amounts of energy on which these taxes are applied, the tax component falls from 4.88 ct/kWh in the Base case to 4.08 ct/kWh under Partial and Full Prosumership use cases. A second cause of the decline is the grid cost. The labor cost component of grid usage cost depends on the amount of energy consumed from the grid. As a result, the Base and No Prosumership use cases pay the highest grid cost (3.94 ct/kWh) while the cases with self-consumption reduce these costs by 22% to 3.11 ct/kWh. While a cost reduction of 5% is only a small incentive to join such an energy community, another benefit of the described setup is price stability. Since about a third of the energy is consumed locally, this fraction of the energy supply is independent of volatility in market prices and energy imports.

### 5.7. Economic Viability: NPV of REC-OC

While RECs offer multitudes of benefits discussed previously, like higher self-sufficiency, lower energy bills for consumers and GHG emissions avoided, it is pertinent to see if investing in RECs is an attractive proposition for investors/lenders. Figure 8 shows the NPV for REC-OC, for setting up and operating REC Pfaff, under different forms of prosumership. When all benefits of prosumership are passed onto the prosumers in the form of lower energy bills (green bars), the NPV of the REC is negative under all cases of prosumership, ranging from −3.0 M€ under Full Prosumership with an annual revenue between 276 k€ and 407 k€ to −0.6 M€ under No Prosumership with annual revenue between 303 k€ and 385 k€. This is because when all cost savings are passed onto prosumers, the overall revenues from operations of REC-OC and the applicable incentives (feed-in and tenant el. incentives) are not sufficient to recoup the initial investment within the project lifetime of 25 years. Alternatively, when the cost savings are not initially passed onto prosumers and are used to repay the loan faster (orange bars), the NPV of the project remains negative but within a smaller range: −1.2 M€ under Full Prosumership use cases, −2.5 M€ under Partial Prosumership and −0.5 M€ under No Prosumership.

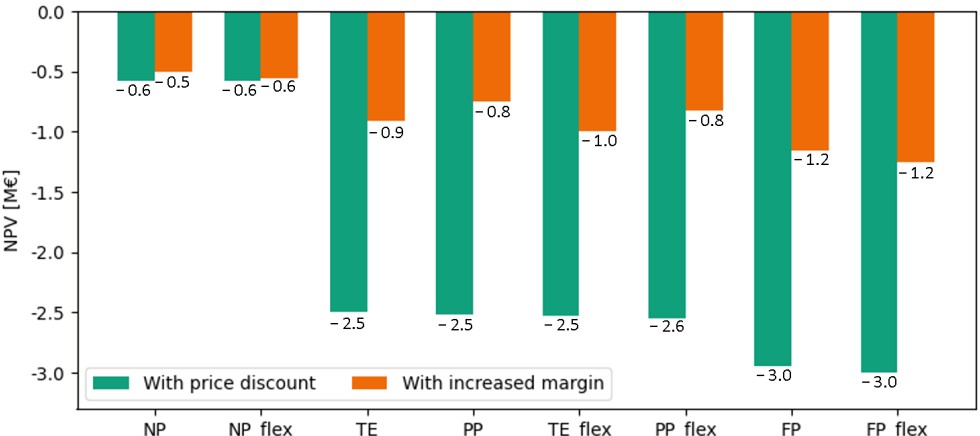

**Figure 8.** NPV of REC-OC after 25 years of operation for different scenarios. Green bars show the NPV with all benefits passed down to the consumer reducing the individual energy bill. Orange bars show the NPV with no consumer price discount but instead using the additional earnings to repay the loan.

One way for the project to break even within 25 years is by decreasing the variable supply costs charged to consumers by at least 20% (3.62 ct/kWh), specifically under Full Prosumership. The price decrease required to break even under other forms of Prosumership would be lower. Currently, RECs may be successful in attracting the interest of investors and can be economically viable through price decreases for consumers but may not be favored by tenants as they do not realize the immediate benefits of becoming prosumers and may even have to bear higher energy costs until the project breaks even.

## 6. Discussion

### 6.1. Price Stability and Economics

For the economic results, we did not take into account any developments in energy prices, inflation rate and CAPEX after January 2022. Since then, energy prices have increased drastically, along with general inflation and costs for PV models. While current developments show the benefits of reducing dependencies on fossil fuels and energy imports boosting the idea of self-producing RECs, a quantitative economic evaluation became harder.

### 6.2. Effect of Demand Side Management

From an energy system view, one benefit of creating RECs is that incentives for local optimization are given. By optimizing self-consumption two beneficial outcomes are expected: 1. a better utilization of RE, 2. a reduction in grid stress. The results show that the first outcome is true for the simulation conducted. However, the effect is small (around 1% increase in SCS) since the shiftable load from EVs in the case study makes up around 5% of the total load. The potential would rise when other flexible devices such as heat pumps or community battery storage systems were introduced. Within the results, we excluded the matter of grid stress reduction. In general, a higher self-consumption leads to less usage of the grid. However, the most important point for grid-friendly behavior is reducing load and production peaks. The EV optimization applied, however, led to an increase in the overall annual load peak by 12%. This was caused by the optimization delaying the EV charging process in the hope of utilizing the PV later, which sometimes led to last-minute charging for many EVs at the same time right before a simultaneous departure of workers within the morning hours. This leads to the statement that self-consumption optimization does not necessarily provide grid stress reduction. Incentives for including peak reduction within the optimization are currently missing.

### 6.3. Regulatory Enablers and Barriers

Insights were gained during the analysis into the regulatory key factors that enable or prevent the foundation of RECs under the RED II and the German regulatory framework and are as follows:

#### 6.3.1. Enablers and Barriers for RECs under the RED II Regulatory Framework

Enablers: The governance structure for RECs defined in RED II firstly recognizes the role of citizens and collective prosumers as important actors in the energy markets. The requirement of heterogeneity of actors in the RECs can prove to be a favorable provision as it creates the foundation for active collaboration between motivated citizens and professional actors such as energy suppliers who are well experienced in the technical and management aspects of energy systems and services. For energy companies, such collaboration can form a future revenue stream while for citizens, it can remove the technical and administrative burden arising from their role as actors in the energy markets. Further, the provision also opens the door to innovative structures and business models which can be utilized for collective prosumership. One such innovation, the RE-CSOP, was discussed in this study; others may follow due to the freedom allowed for this in RED II. The biggest enabler is the enabling framework that the Member States are mandated to develop in their national

laws to facilitate the participation of RECs in the energy market on an equal footing with established actors.

Barriers: According to the provisions of RED II, self-consumption is explicitly encouraged by exempting it from all levies and fees. However, such an incentive is missing for energy sharing, which is subject to sharing the cost of energy systems fairly and proportionately [4]. As a result, in its current form, the RED II encourages either Partial Prosumership or No Prosumership in RECs but is ill-suited towards Full Prosumership. Energy sharing is beneficial to the system operators who can postpone or avoid reinforcing the grid due to reduced feed-in of energy at points of RE generation. By offsetting the benefits of energy sharing against the cost of grid reinforcement, the case can be made for an explicit incentive for energy sharing and Full Prosumership in future revisions of the RED.

### 6.3.2. Barriers and Enablers for RECs under the German Regulatory Framework

Enablers: The Tenant Electricity Model eradicates the barrier that prevented people living or working in a rental unit within a multi-party building from consuming PV energy produced on-site. By offering the Tenant Electricity surcharge, the legislation explicitly incentivizes energy providers to invest in rooftop PV generators and to sell the energy for a discount in the same or neighboring buildings. Other incentives in the form of reduced variable taxes bring down the overall energy bill for consumers and make the Tenant Electricity Model an attractive option to consume energy. The German framework incentivizes PV systems up to 100 kW, a limit much higher than the 30 kW envisioned in RED II. Therefore, the German framework promotes a larger technical potential for prosumership compared to RED II. The 'supply chain model' introduced in the EEG 2021 has removed barriers preventing landlords from participating in Tenant Electricity by allowing them to outsource their obligations as an energy supplier to third parties [13].

Barriers: Certain barriers hinder the Tenant Electricity Model from becoming a standard mode of collective prosumership in Germany. The main drawback is that in the setup, people living or working within a building are still only passive consumers and do not own shares of the PV generator. New PV plants are mainly built by professional Tenant Electricity providers, or built by landlords with the operation outsourced to a professional third party. In the model, a classic energy supply contract for the overall energy supply is offered to the tenants, which comes with all the obligations of an energy supplier. This bureaucratic barrier is high and comes with higher operational costs. Additionally, the Tenant Electricity Model is open only to buildings with 40% area under residential usage, thereby preventing tenant-occupied commercial buildings from participating in this model [13]. Finally, the energy exchange is permitted only from one building to another or ancillary building if it is in the same neighborhood and without the usage of the public grid [13].

### 6.4. Outlook

Future research on the project can focus on several lines of inquiry. The approach developed and applied in this article can be replicated in other member states where provisions of RED II related to RECs have fully or partially transposed into national laws, such as in Belgium (Flanders), Italy, France, Austria, Ireland, Denmark, Lithuania, Spain and Portugal [43]. Research questions also arise with regard to the mechanisms of distributing and allocating costs and profits generated by the REC, e.g., the ratios between equity and loan, exemptions for low-income tenants, the manner of sharing dividends with the community members and suitable methods of re-investing earnings generated from the REC back into the community. This would extend the knowledge and experience on the functioning of RECs and thus drive their uptake.

### 7. Conclusions

In this paper, we investigate an existing neighborhood in Germany to evaluate how it can and should be organized as a Renewable Energy Community (REC). We look at compliance with the Renewable Energy Directive (RED II) as well as the ecological and

economic implications of establishing a REC. Derived from our quantitative simulation results, we discuss which elements of the current EU and German regulation act as enablers or barriers for RECs. While our literature research demonstrated that a wide array of studies on specific RECs as well as papers looking at the regulatory aspects of RECs, there is a lack of studies looking at RECs from a holistic perspective involving their regulatory, organizational, technical, ecological and economic aspects simultaneously. The novelty of this paper is its contribution to closing this gap by undertaking an integral, multi-disciplinary assessment of a potential REC.

The study contributes to existing knowledge firstly by organizing and financing a REC that can be set up in a tenant-occupied, mixed-use urban district with a methodology that can be applied to similar neighborhoods. This is achieved using the emerging concept of Renewable Energy Consumer Stock Ownership Plans (RE-CSOPs), which are compliant with RED II and offer additional benefits over existing organizational forms of community energy initiatives, and therefore may become favored organizational structures for RECs across the EU compared to the traditional forms such as cooperatives.

Secondly, the study classifies collective prosumership into various use cases based on the activities pursued and differentiates the impact of regulatory provisions based on the degree of prosumership pursued.

Thirdly, we use a single, replicable model to assess the technical feasibility, ecological benefits and economic viability of pursuing prosumership through RECs. We find that collective prosumership through RECs offers a multitude of benefits to the citizens, local energy systems and the environment and the extent of these benefits varies with the type of prosumership pursued.

When compared to the Base Case, Full Prosumership (involving self-consumption and sharing of energy within the REC) offers the highest self-sufficiency (SSS of 35%) and GHG reduction of 35%, followed by Partial Prosumership (where the PV energy is self-consumed and the excess is fed to the grid), which offers an SSS of 28% and 29% GHG avoided. The lowest benefits are accrued from No Prosumership (where PV is generated only to be supplied to the grid). The benefits of RECs are not only limited to the prosumers but other stakeholders as well, e.g., by opening a new revenue stream for energy utility as an operator and service provider for RECs .

Lastly, we find that although Partial and Full Prosumership lead to energy bill savings of 5% for consumers compared to the Base Case, we find that the NPV for all use cases would be negative, leading to the conclusion that such projects will face challenges in attracting investments. A financial boost of 3.6 ct/kWh was found to be needed to become profitable. Anyway, RECs are envisioned as entities whose primary purpose is to provide environmental, economic or social community benefits for members or for the local areas where it operates, rather than financial profits [4].

The EU and German regulatory frameworks were found to have several enablers that favor collective prosumership via RECs but are bogged down by several barriers. On the EU level, the RED II in its current form encourages self-consumption, whereas energy sharing is not defined. Establishing energy sharing would, however, allow future RECs to pursue Full Prosumership and realize its maximum benefits, which are higher than Partial Prosumership as shown by this study. At the German national level, the Tenant Electricity Model offers incentives to residential consumers to consume local green energy and save money simultaneously and has removed barriers that previously discouraged landlords from participating. However, by limiting participation to residential-dominated buildings and by forbidding the usage of the public grid when exchanging energy, the model discourages the formation of scalable RECs in mixed-use neighborhoods consisting of several buildings.

**Author Contributions:** Conceptualization, S.C. and A.S.; Data curation, A.S.; Formal analysis, M.K. and F.P.; Funding acquisition, A.S. and M.K.; Investigation, S.C. and A.S.; Methodology, S.C. and A.S.; Project administration, A.S.; Resources, A.S.; Software, A.S.; Supervision, A.S.; Validation, S.C. and A.S.; Visualization, S.C. and A.S.; Writing—original draft, S.C. and A.S.; Writing—review & editing, M.K. and F.P. All authors have read and agreed to the published version of the manuscript.

**Funding:** The work presented has been conducted in the EnStadt:Pfaff project. The research leading to these results has received funding from the German Ministry of Education and Research (BMBF) under the funding number 03SBE112G and the Ministry for Economics and Energy (BMWi) under the funding number 03SBE112D.

**Institutional Review Board Statement:** Not applicable.

**Informed Consent Statement:** Not applicable.

**Data Availability Statement:** Data will be made available upon request.

**Acknowledgments:** The research scope of this study was developed within the Fraunhofer Cluster of Excellence Integrated Energy Systems (CINES).

**Conflicts of Interest:** The authors declare no conflict of interest. The funders had no role in the design of the study; in the collection, analyses, or interpretation of data; in the writing of the manuscript; or in the decision to publish the results.

## Abbreviations

The following abbreviations are used in this manuscript:

| | |
|---|---|
| $CO_2$ eq | carbon dioxide equivalents |
| CF | Cashflow |
| CSOP | Consumer Stock Ownership Plan |
| CEC | Citizen Energy Community |
| EEG | Erneuerbare Energien Gesetz |
| EU | European Union |
| FP | Full Prosumership Scenario |
| GHG | Greenhouse Gas |
| KPI | Key Performance Indicator |
| kW | kilowatt |
| kWh | kilowatt-hour |
| NP | No Prosumership Scenario |
| NPV | Net Present Value |
| PP | Partial Prosumership Scenario |
| PV | Photovoltaic |
| RE | Renewable Energy |
| REC | Renewable Energy Community |
| REC-OC | REC (Pfaff) Operating Company |
| RED | Renewable Energy Directive |
| RES | Renewable Energy Sources |
| RQ | Research Question |
| SCS | Self-Consumption Share |
| SME | Small and Medium Enterprise |
| SSS | Self-Sufficiency Share |
| TE | Tenant Electricity |

## Appendix A

**Table A1.** Electricity Tariffs Applied to Prosumership Use Cases.

| Parameter | Unit | Grid-Supplied | Self-Consumed | Shared | Tenant Electricity | Source |
|---|---|---|---|---|---|---|
| **Fixed Supply Tariff** | | | | | | [44] |
| Residential | $\frac{€}{a}$ | 118.2 | 118.2 | 118.2 | 106.38 | |
| Commercial ($\leq10\frac{MWh}{a}$) | $\frac{€}{a}$ | 118.2 | 118.2 | 118.2 | 106.38 | |
| Commercial ($>10\frac{MWh}{a}$) | $\frac{€}{a}$ | 0 | 0 | 0 | 0 | |
| **Variable Supply Tariff** | | | | | | [44] |
| Residential | $\frac{ct}{kWh}$ | 20.19 | 20.19 | 20.19 | 18.17 | |
| Commercial ($\leq10\frac{MWh}{a}$) | $\frac{ct}{kWh}$ | 21.21 | 21.21 | 21.21 | 19.09 | |
| Commercial ($>10\frac{MWh}{a}$) | $\frac{ct}{kWh}$ | 22.14 | 22.14 | 22.14 | 19.93 | |
| **Fixed Retailer Margin** | | | | | | Author calculation |
| Residential | $\frac{€}{a}$ | 22.46 | 22.46 | 22.46 | 20.21 | |
| Commercial ($\leq10\frac{MWh}{a}$) | $\frac{€}{a}$ | 22.46 | 22.46 | 22.46 | 20.21 | |
| Commercial ($>10\frac{MWh}{a}$) | $\frac{€}{a}$ | 0 | 0 | 0 | 0 | |
| **Variable retailer Margin** | | | | | | Author calculation |
| Residential | $\frac{ct}{kWh}$ | 4.75 | 4.75 | 4.75 | 4.28 | |
| Commercial ($\leq10\frac{MWh}{a}$) | $\frac{ct}{kWh}$ | 4.94 | 4.94 | 4.94 | 4.45 | |
| Commercial ($>10\frac{MWh}{a}$) | $\frac{ct}{kWh}$ | 5.12 | 5.12 | 5.12 | 4.61 | |
| **Fixed Grid Usage Tariff (Low Voltage)** | | | | | | [45] |
| Performance price ($\leq2500\frac{h}{a}$) | $\frac{€}{kWa}$ | 26 | 26 | 26 | 26 | |
| Performance price ($>2500\frac{h}{a}$) | $\frac{€}{kWa}$ | 94.3 | 94.3 | 94.3 | 94.3 | |
| Labour price ($\leq2500\frac{h}{a}$) | $\frac{ct}{kWh}$ | 5.37 | 5.37 | 5.37 | 5.37 | |
| Labour price ($>2500\frac{h}{a}$) | $\frac{ct}{kWh}$ | 2.63 | 2.63 | 2.63 | 2.63 | |
| **Variable Electricity Tax** | | | | | | |
| Electricity Tax | $\frac{ct}{kWh}$ | 2.05 | 2.05 | 2.05 | 2.05 | [44] |
| KWKG Surcharge | $\frac{ct}{kWh}$ | 0.378 | 0 | 0.378 | 0 | [4,13,44] |
| EEG Surcharge | $\frac{ct}{kWh}$ | 0 | 0 | 0 | 0 | [4,13,44] |
| Strom NEV Surcharge | $\frac{ct}{kWh}$ | 0.358 | 0 | 0.358 | 0 | [4,13,44] |
| EnWG Surcharge | $\frac{ct}{kWh}$ | 0.419 | 0 | 0.419 | 0 | [4,13,44] |
| AbLAV surcharge | $\frac{ct}{kWh}$ | 0.003 | 0 | 0.003 | 0 | [4,13,44] |
| Concession fees | $\frac{ct}{kWh}$ | 1.59 | 0 | 1.59 | 0 | [46] |
| Total var. el. tax | $\frac{ct}{kWh}$ | 4.798 | 2.05 | 4.798 | 2.05 | Author calculation |
| **Total prices for end users** | | | | | | Author calculation |
| Res. Annual fee | $\frac{€}{a}$ | 140.66 | 140.66 | 140.66 | 126.59 | |
| Res. Consumption based | $\frac{ct}{kWh}$ | 29.74 | 26.99 | 29.74 | 25.5 | |
| Commercial ($\leq10\frac{MWh}{a}$) Fixed price | $\frac{€}{a}$ | 140.66 | 140.66 | 140.66 | 126.59 | |
| Commercial ($\leq10\frac{MWh}{a}$) Cons. based | $\frac{ct}{kWh}$ | 30.95 | 28.2 | 30.95 | 25.59 | |
| Commercial ($>10\frac{MWh}{a}$) Fixed price | $\frac{€}{a}$ | 0 | 0 | 0 | 0 | |
| Commercial ($>10\frac{MWh}{a}$) Cons. based | $\frac{ct}{kWh}$ | 32.06 | 29.31 | 32.06 | 26.59 | |

**Table A2.** Incentives [in ct/kWh] [42].

| | <10 kWp | <40 kWp | <100 kWp | <400 kWp | <1 MWp |
|---|---|---|---|---|---|
| Feed-in tariff (Full Feed-in) | 13.4 | 11.3 | 11.3 | 9.4 | 8.1 |
| Feed-in tariff (Excess Feed-in) | 8.6 | 7.5 | 6.2 | 6.2 | 6.2 |
| Tenant el. incentive | 3.79 | 3.52 | 2.37 | - | - |

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
