# Peer review of "Renewable Energy Communities as Modes of Collective Prosumership: A Multi-Disciplinary Assessment Part II—Case Study"

_energies, doi:10.3390/en15238936_

Round 1

Reviewer 1 Report

To start with, I would like to thank authors for their work in terms of interesting topic, high quality research and well written article(s) and want to mentioned the research conducted by authors is described in two articles energies-2049923 and energies-2049926.

The research is devoted to the development of integrated model to assess potential Renewable Energy Communities in the Member States of the European Union through analyzing the energy, ecological and economic impacts for the buildings and for Renewable Energy Communities.

Overall, the research is at high level and I have no any concerns to it at all (small typos will be eliminated by language editors). However, I see one principal problem. The research is as voluminous enough so authors decided to divide it into two articles the current one (energies-2049923) and energies-2049926. The decision caused the following problem:

The first paper (energies-2049923) contains only the theoretical results of the Model building without the example of model application \ the comparison with others through the case study. While the second paper (energies-2049926) provides the case study of the model application in Germany on the theoretical research presented in the first one. So, without each other they can not be considered as of full value. However, readers of one paper can be not aware of the existing of other one and miss it. I cannot see the solution to cite papers in each other, except for the publication of the second paper after the first one. But, anyway, the problem will be half ameliorated.

As papers describe one research and they must have the accepted article structure, some information is doubled and can be considered as plagiarism. Please, remember your statement in cover letter – “We confirm that neither the manuscript nor any parts of its content are currently under consideration or published”

I recommend authors to redesign the whole research in one paper or solve the problem with editors.

 The same review will be for the both papers (energies-2049923 and energies-2049926.)

Author Response

Please see the attachment. Note that all line numbers refer to the revised document.

Reviewer 2 Report

The authors investigate an existing neighbourhood in Germany to evaluate how it can and should be organized as a Renewable Energy Community.

Please, make the following corrections:

-         provide a zoom of Table 1; 

-         correct the typo in row 226;

-         provide a zoom of labels and legend in Figures 3 and 6.

Author Response

Dear reviewer, thank you for your feedback. Concerning English proofreading we gave the paper to a native colleague but did not get feedback yet. However, we will work in the feedback as soon as possible. In addition, we increased zoom of Table 1. We originally set it to be a tiny table so it could fit in the text width of the paper, but since full page length tables are also possible within the layout, readability is now enhanced. Also, the texts within figures 3 and 6 were increased. Concerning the typo in row 226. This will be addressed shortly before publication. We reached out to the editorial since the source we want to mention here is not yet published. So, the current typo is a place holder for the correct citation to be included.

Round 2

Reviewer 1 Report

Dear authors,

Thank you for their huge work spent on the revision and addressing comments of mine and other reviewers!

I have already told that I liked the paper in terms of the quality research and the paper design. Concerning to merging both papers, I fully agree that purposes of each paper are different and might be separated. I am really glad that you can have negotiated this question with editorial office and you have added additional information on mentioning each paper.

I recommend your paper for publication and wish you continue working on such qualitive research